# New Reliability Studies of Data-Driven Aircraft Trajectory Prediction

**Seyed Mohammad Hashemi** [1,†]**, Ruxandra Mihaela Botez** [1,*] **and Teodor Lucian Grigorie** [2,†]

1   ÉTS, LARCASE, 1100 Notre Dame West, Montreal, QC H3C 1K3, Canada;
    seyed-mohammad.hashemi.1@ens.etsmtl.ca
2   Military Technical Academy Ferdinand I, 39-49 George Cosbuc, 040531 Bucharest, Romania;
    ltgrigorie@yahoo.com
*   Correspondence: ruxandra.botez@etsmtl.ca
†   These authors contributed equally to this work.

**Abstract:** Two main factors, including regression accuracy and adversarial attack robustness, of six trajectory prediction models are measured in this paper using the traffic flow management system (TFMS) public dataset of fixed-wing aircraft trajectories in a specific route provided by the Federal Aviation Administration. Six data-driven regressors with their desired architectures, from basic conventional to advanced deep learning, are explored in terms of the accuracy and reliability of their predicted trajectories. The main contribution of the paper is that the existence of adversarial samples was characterized for an aircraft trajectory problem, which is recast as a regression task in this paper. In other words, although data-driven algorithms are currently the best regressors, it is shown that they can be attacked by adversarial samples. Adversarial samples are similar to training samples; however, they can cause finely trained regressors to make incorrect predictions, which poses a security concern for learning-based trajectory prediction algorithms. It is shown that although deep-learning-based algorithms (e.g., long short-term memory (LSTM)) have higher regression accuracy with respect to conventional classifiers (e.g., support vector regression (SVR)), they are more sensitive to crafted states, which can be carefully manipulated even to redirect their predicted states towards incorrect states. This fact poses a real security issue for aircraft as adversarial attacks can result in intentional and purposely designed collisions of built-in systems that can include any type of learning-based trajectory predictor.

**Keywords:** aircraft trajectory prediction; deep neural network; reliability; adversarial attack

## 1. Introduction

Avionics transportation standards and policies established by official agencies require all aviation companies to respect the approved safety protocols. These standards have been developed to ensure safe aircraft transportation, especially for modern automatic flights. Huge investments have been made in the United States over the last decades by the Federal Aviation Administration (FAA) into "The Next-Generation of Aerial Transportation" project, with the aim of increasing the safety and reliability of flights [1].

Safety protocols are required for air traffic control, safe path definition, and collision avoidance, which determine conditions in which aircraft are allowed to fly, while safety policies reduce the chance of collisions. In this way, aircraft trajectory prediction (ATP) can be considered as an excellent tool for achieving safe aerial transportation. This prediction method may be used at different times, including for short-term and long-term predictions. Long-term prediction is useful for air traffic control, fuel consumption optimization, and logistics operations while short-term prediction is useful for conflict

detection. The predicted trajectories may be utilized by ground computer units as part of an air traffic control system (ATCS) or by computer units of the aerial collision avoidance system (ACAS) in the cockpit.

Many aerial control tasks are processed by avionics systems. Such tasks might include aircraft trajectory optimization [2] and its application into flight management systems [3], which aim to reduce operational costs [4], fuel consumption, and adverse environmental side effects [5]. A variety of algorithms, such as genetic algorithm (GA) [6], particle swarm optimization (PSO) [7], ant colony [8], bee colony [9], beam search [10], and harmony search [11] have been employed to solve aircraft trajectory optimization problems. However, the main aim of avionics control systems is aerial collision avoidance [12], where ATP contributes to solving encounter scenarios efficiently. This paper is focused on ATP accuracy and reliability; the accuracy of predicted trajectories was assessed using the error rate in the test phase, and the reliability of ATP neural network models was evaluated based on the fooling rate for the adversarial attack. Evaluating the ACAS performance analysis based on data-driven trajectory predictors was not the aim of this article.

Generally, the predicted trajectory for each moving aircraft, produced by an algorithm, consists of a sequence of position states in the Cartesian space with their respective displacements from other sequences needed to prevent aircraft from colliding with each other. In cases when a safe zone constraint related to the predicted paths is violated, real-time adjustment is required from the prediction system in order to rearrange the aircraft position states [13]. In this type of setup, the computational complexity of the predictors is a key factor in providing a rapid and practical solution [14] as delays in aircraft equipped with aircraft trajectory prediction (ATP) systems can result in costly and mainly dangerous collisions. In this paper, a novel algorithm is proposed for real-time and accurate ATP in order to meet the high standards of a reliable control system.

Among all the algorithms developed for ATP, neural networks, especially deep learning approaches, have shown the most accurate performance if enough training data are provided. Many public trajectory datasets that are available online can be used for this aim. Deep learning (DL) models trained for path prediction purposes significantly outperform any other data-driven algorithms based on comparisons of runtime, from regression correctness to computational complexity. Unfortunately, recent studies have uncovered the vulnerability of all data-driven models, whereby some input samples can be purposely manipulated to mislead them [15]. These fake samples are known as adversarial samples and, unfortunately, detection of fake sample intrusion is presently an ongoing problem for the machine learning community. In this paper, the existence and impacts of these samples are characterized in relation to ATP for both conventional regressors and cutting-edge deep learning models.

The organization of this paper is as follows. The common approaches developed for air vehicle trajectory prediction are reviewed in the following section. Brief explanations of data-driven predictors are provided in the third section. Section four is dedicated to our experimental results and to a deep analysis of adversarial attacks on a variety of trained models. Finally, the related ongoing problems are elaborated for future works. For instance, we can characterize the existence of adversarial attacks for any learning-based algorithm while there is no certain systematic defense. Moreover, unfortunately, studies show that these adversarial samples are transferable from one model to another, even if they have been manipulated for other algorithms.

## 2. Related Works on Trajectory-Based Operations

Generally, trajectory-based operations are categorized as either short- or long-term predictions, whereby each prediction type has its own advantages relevant to the corresponding task. Figure 1 depicts the general setup of an aircraft encounter scenario, which could be visualized in short- and/or long-term prediction frameworks. Although encounter scenarios, such as the one shown in Figure 1, have been solved using Traffic Collision Avoidance System (TCAS) without future trajectories, the collision avoidance task can be performed more optimally by relying on predicted trajectories. In fact, it is known that the TCAS modifies the owner's future trajectory if an intruder enters into the

owner's resolution advisory zone. Moreover, the TCAS design is based on the current aircraft position and on its conservative virtual unsafe zone. Hence, many false alarms and unnecessary resolution advisory events may occur during the flight. In this way, the collision zone can be reduced using an accurately predicted trajectory (position of aircraft in the nth step), which leads to avoiding unnecessary trajectory modification. Therefore, the design of a reliable and precise trajectory prediction algorithm is needed [16].

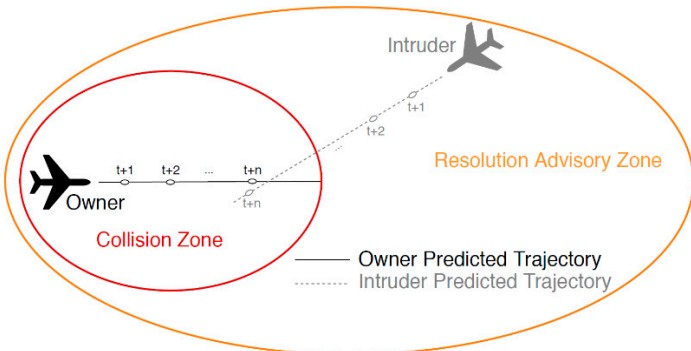

**Figure 1.** Encounter scenario.

There is a large volume of research targeting these frameworks. Since the trajectory prediction could be formulated as a regression problem, these researches could be mainly employed for improving regression performance [17]. In all these frameworks, the encounter scenario is defined based on owner and intruder attitudes [16]. The encounter scenario may occur due to the pilot mistake [18], lack of visibility [19], actuator failure [20], or loss of communications [21]. When an intruder arrives in the vicinity of the owner's neighborhood, after intruder detection [22], the ACAS resolution advisories commands should be transferred to the fixed-wing aircraft control system, which is supposed to deflect control surfaces with the aim to modify future trajectory. The control system of the owner aircraft (that is flying in a specific route) updates its subsequent trajectory with respect to the built-in regression model, while sensory radar information is being provided simultaneously [23]. Finally, the safety control system takes proper actions in order to avoid a possible collision [24].

*2.1. Collision Avoidance*

Model predictive control is an algorithm designed for trajectory prediction and path planning [25,26]. This strategy is used to model both the dynamics and the kinematics of a moving vehicle in order to predict the most appropriate trajectory to be followed. In contrast to this deterministic approach, a stochastic method is proposed in [27], which implements the assembly of the multiple models to be tuned via optimization techniques. Since real-time optimization for prediction, even for a single model, is very costly, an adaptive control model that runs quadratic programming optimizers is developed in [28].

In [29], a nonlinear model predictive setup was proposed in an effort to solve multiconvex obstacles. A linear optimization algorithm was designed for the owner aircraft model to avoid collisions with other aircraft models. A multiagent control policy for handling complex encounter scenarios is discussed in [30]. Since the agents were distributed, and the agreement of each agent was needed, the optimization problem was expensive. Instead, their optimization procedure generated more accurate position states that were followed by the aircraft.

Given that nonlinearities and uncertainties are involved in all these optimization problems, research efforts have been employed to develop evolutionary algorithms for fine path regression. Particle swarm optimization (PSO) has been adapted to generate 3D position states illustrated by B-spline curves [31]. Genetic algorithm (GA) is a greedy-based evolutionary procedure that has been utilized for greedy regression in highly convex scenarios [32]. This algorithm incorporates only

local nearby consistencies in its optimization routine with the aim of generating to-the-point states regardless of the entire path. In order to fully take advantage of the benefits of GA and PSO approaches, their combination has been proposed in [33], and it was demonstrated that the combined GA–PSO algorithm was able to outperform GA and PSO individually.

In addition to these greedy approaches, data-driven algorithms have been developed for trajectory predictions by rectifying the trajectories' local state shortcomings. For example, a neural-network-based clustering approach that implements an unsupervised learning process is discussed in [34]. In some research studies, deep neural networks have been utilized for two aims, firstly for safe zone clustering and, secondly, for correct position prediction of an aircraft over time [35,36]. Deep reinforcement learning approaches have also been embedded into this setup, and promising results have been reported thus far [37].

### 2.2. Data-Driven Trajectory Prediction

A long-term aircraft trajectory is predicted using a trained hidden Markov model (HMM) [38] using 3D positional and, in addition, environmental data, which are considered as the fourth dimension of the dataset needed to consider weather uncertainties. That work divides the whole path into small patches of 3D cubes and then predicts the future trajectory under real flight conditions. Similarly, a long-term four-dimensional (4D) aircraft trajectory has been predicted using a deep generative neural network architecture modeled in the presence of uncertainties, such as wind, convective weather, and temperature [39].

In [40], aircraft trajectory prediction is considered as a flight sequence estimation problem. That work proposes a recurrent neural network for trajectory prediction. The results reveal noticeable improvements in state predictions. Following this idea, a long short-term memory (LSTM) algorithm has been developed which outperformed its predecessor [41]. Although a comparative study conducted in [42] showed that deep learning algorithms outperform all other machine learning approaches, a variety of their models were implemented to further investigate their prediction capability, as well as their vulnerability to adversarial attacks.

Overall, the trajectory may be predicted using conventional approaches (e.g., model predictive control (MPC)) based on aircraft dynamics models or modern data-driven techniques (e.g., deep neural network (DNN)) that rely on large amounts of recorded data. Studies have shown that modern data-driven techniques outperform conventional approaches if enough training data is available and security issues are respected. It is known that in conventional approaches, uncertainties backpropagate through the prediction horizon, and errors increase dramatically. Hence, the data-driven algorithms were adopted for trajectory prediction, and the ATP task was performed regardless of aircraft dynamics models, which is a remarkable advantage of data-driven predictors. A carefully tuned and real-time predictable path is therefore required for each aircraft. Since data-driven algorithms have been used for path prediction, they have been found to be not completely fault-tolerant, and they may create security issues for aviation transportation systems. In the following section, some of our benchmarking algorithms and datasets are explained. Then, we will explain how the adversarial samples can be generalized to models being trained using standard aircraft trajectory datasets.

## 3. Building Data-Driven Predictors

Data-driven predictors have shown great performance in all regression tasks, which is also shown in our present study. Therefore, several different learning-based algorithms are explored for solving the aircraft trajectory regression problem (ATRP). The benchmarking algorithms that we will propose range from conventional (e.g., logistic regression) to state-of-the-art (e.g., convolutional neural network). The performance of these algorithms is totally dependent on the characteristics of the given dataset and on its sample distributions, in which sampling distribution is defined as a probability distribution of a statistic that is derived from a considered population. Since there is no practical approach to define the best regression algorithm for our dataset, conducting experiments on all of them

to determine the most proper one is needed. Although, nowadays, deep learning-based approaches (such as CNN, LSTM) are the best performing algorithms, there is no guarantee of outperforming conventional algorithms, such as support vector regression (SVR). Due to these reasons, six regression algorithms have been included in our study. These regression algorithms are logistic regression (LR), support vector regression (SVR), deep neural network (DNN), convolutional neural network (CNN), recurrent CNN (RNN) and, finally, long short-term memory (LSTM). Our motivation for utilizing all these algorithms is to measure and compare the strength, generalizability, and robustness of these models. Brief explanations are provided for each of these algorithms in the following subsections.

*3.1. Logistic Regression (LR)*

Logistic regression has the potential to fit its results to the training data if the uniformity of the given dataset is standard and without fluctuations. Since our experimental dataset is "evenly" distributed over time, it does not contain noticeable fluctuations and, thus, LR can learn from the mentioned dataset finely and make accurate predictions [43].

By assuming that the given inputs and outputs to the algorithm are $X$ and $Y$, respectively, Equation (1) is considered for LR model learning [44]:

$$Y = \frac{1}{\exp(\theta_0 - \theta_1 x_1 - \theta_2 x_2 - \cdots - \theta_n x_n)} \tag{1}$$

where $\theta$ is the weight vector that could be obtained during training by optimization using a relevant cost function $J(\theta)$. Conventionally, the cost function is defined in Equation (2) [45]

$$J(\theta) = \frac{1}{m} \sum_{i=1}^{m} [-y_i \log(h_\theta(x_i)) - (1 - y_i) \log(1 - h_\theta(x_i))] \tag{2}$$

where the number of samples is denoted by m in the training set, and $h_\theta(X)$, known as the hypothesis, is defined in Equation (3) [46]

$$h_\theta(X) = \frac{1}{\exp(-\theta X^T)} \tag{3}$$

where $\theta \in \{\theta_i\}$. One of the crucial observations is that the logistic function $\theta$ considered in the above equations increases the risk of saturation during the training phase; the regularization term is added, as shown in Equation (4), to rectify this problem [45]:

$$\frac{\lambda}{2m} \sum_{j=1}^{n} (\theta_j)^2 \tag{4}$$

where $\lambda$ the regularization term that binds the cost function given by Equation (3) to more parameters shown in Equation (4) in order to improve the model's precision. $\lambda$ should be manually tuned with respect to the training statistics. Training statistics refer to weight vectors obtained while running an iterative process for learning, in which their fine tuning increases the chance of obtaining better weight vectors. The addition of this term to Equation (2) contributes to avoiding overfitting of the dataset and the need to memorize samples. We trained this LR model on some standard aircraft trajectory datasets and fine-tuned its hyperparameters. The basic problem of this regressor is its generalization to complicated patterns, which could be challenging for the LR model to learn. Hence, support vectors are used to capture data distribution better than the LR, even in cases when the training data are not linearly separable.

### 3.2. Support Vector Regression (SVR)

This conventional regressor is based on the well-known principle of support vector machines, which is capable of learning from high-dimensional spaces. The concept supported by the SVR is the mapping of training data from the Euclidean space to another higher dimension space by using the "kernel trick", then the learning of the decision boundaries. There are many kernel functions that could implement this mapping, such as homogeneous/inhomogeneous polynomials, tan h, Gaussian, and others. Different experiments were performed by us on these kernels in order to determine and adopt the best ones. The optimization process employed for our SVR model is given in Equation (5) [47]:

$$\min \frac{1}{2} \| \theta \|^2 \ s.t. \begin{cases} \theta_i x_i + b - y_i \leq \epsilon \\ y_i - \theta_i x_i - b \leq \epsilon \end{cases} \tag{5}$$

where $\epsilon$ denotes the decision boundary precision which should be tuned carefully. The performance of the SVR model as well as the performance of any other learning-based algorithm is totally dependent on the type of regression task and on the dataset used for training. Moreover, SVR learns from a mapped subspace, which could be very challenging. To address this potential problem, some other algorithms that can learn from raw samples are used. The state-of-the-art of these algorithms will be reviewed in the following subsections.

### 3.3. Deep Neural Network (DNN)

Neural network algorithms have been implemented for many regression tasks. It has been shown that multilayer perceptrons (MLPs) can produce accurate models for any regression problem if enough training samples are provided [48]. With the advancement of deep neural networks, many interesting architectures have been introduced, outperforming MLPs. These algorithms learn from raw data, and can be used to solve time series problems [49], such as aircraft trajectory prediction.

Unlike conventional data-driven models, modern DNNs learn from training sample distributions with any dimensionality; sometimes, dimension conversion has to be conducted with respect to the complexity of the regression task. This fact means that learning-based algorithms can be categorized into feature-based (conventional algorithms such as SVR, LR) and raw inputs (modern deep learning algorithms, such as CNN, LSTM). The latter category does not need to be provided by handcrafted feature vectors, but they need more training samples than conventional algorithms. Otherwise, their performance may decrease. It is important to have a large enough dataset for training deep learning algorithms. When there is no access to a large dataset, transferring of dimensions can be applied to enhance sample distributions in order to improve the algorithm performance. To some extent, DNNs can be sensitive to the volume of the training set, and their performance may degrade if the training dataset is not large enough. To rectify this issue, several data augmentation algorithms have been proposed [50].

Similarly to MLPs, input, hidden, and output layers are the mains components of DNNs. New proposed architectures for DNNs include very dense hidden layers with a massive number of filters. AlexNet [51], GoogLeNet [52], and ResNet [53] are some of the modern architectures proposed for DNNs.

Cutting-edge DNN architectures consist of very deep hidden layers, but they also take advantage of modern blocks in their hidden layers, such as dropout [54], rectified nonlinear activation functions [55], and optimized cost functions with momentum and adaptive learning rates [56]. "Dropout" is a regularization technique for training a neural network. It randomly freezes some weight vectors in the training process and avoids updating them to the end of the ongoing epoch, which boosts the training performance especially for very dense CNNs. Rectified nonlinear activation function is a discrete activation function including two linear functions. Mathematically, $ReLu\ (x) = \begin{cases} 0 \ for \ x \leq 0 \\ x \ for \ x > 0 \end{cases}$ . It has been demonstrated that it outperforms the traditional sigmoid function in neural network training.

Momentum and adaptive learning rate tune the training cost function with slightly perturbation weight vectors toward the maximum variations direction.

The abovementioned DNN architectures have been developed for complex computer vision applications, and they are not fully compatible with the aircraft trajectory regression problem (ATRP). Therefore, we propose our DNN architecture, adapted to our dataset as shown in Figure 2.

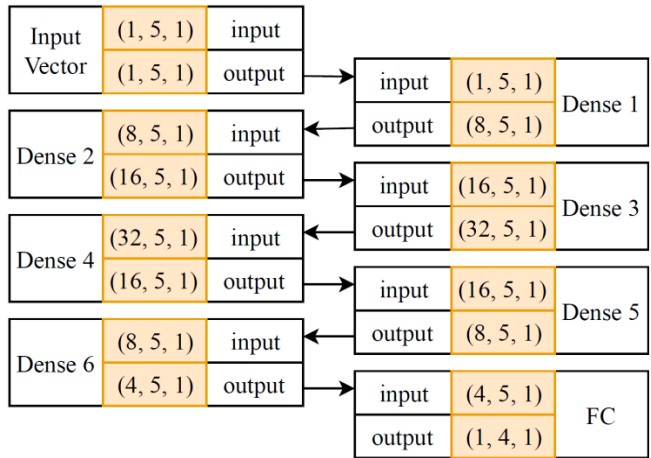

**Figure 2.** Proposed deep neural network (DNN) architecture for the aircraft trajectory regression problem (ATRP).

This figure shows the architecture of our proposed DNN. There are three types of blocks, namely the input vector (input layer), dense (hidden layer), and fully connected (FC). The dense and FC layers are the same, but the latter is not connected to any other layers after it. The highlighted parts of each block specify its input and output dimensions as well as the number of trained filters (weight vectors per layer) shown in triplet of (number of filters, input dimensions, output dimensions). For instance, by considering the input of Dense 2, triplet of (8,5,1) means that there are eight filters in this layer, and the dimension of the input vector is $5 \times 1$. Moreover, the connection between layers is shown with oriented arrows.

The input layer in this architecture is a $1 \times 5 \times 1$ tuple consisting of one filter. Filter dimension is defined based on the input dataset which consists of five recorded measured parameters represented as $[\text{latitude, longitude, altitude, velocity, time}]^T_{5 \times 1}$. "Hidden layers" are shown as five dense layers that are fully connected to the next layer in order to produce outputs as $[\text{latitude, longitude, altitude, time}]^T_{4 \times 1}$. Except for the latter layer, all other layers include batch normalization [57] with normal distribution, dropout with a 0.5 ratio, and rectified linear unit ($ReLU$) activation function as a nonlinearity.

Unfortunately, there is no deterministic approach for designing DNN architecture that is obtained after running several exploratory experiments and after achieving the desired DNN performance in terms of regression accuracy or error rate. In order to avoid overfitting our model to the training set, "early stopping" [58] was used to achieve the highest regression accuracy while keeping it still generalizable.

Although DNN filters are capable of learning very complex sample distributions, incorporating convolution layers can noticeably improve model performance. These layers could be added with/without dense hidden layers, and they could be revealed by running experiments on the given dataset. In the next subsection, our desired CNN architecture for the ATRP is presented.

### 3.4. Convolutional Neural Network (CNN)

Assuming that a random function $g_i(\theta)$ and an input sample $x_i \in \{X\}$ are given, then their convolution, $g_i \circledast x_i$ for $i \in \{1, 2, 3, \ldots, n\}$ will give a "convolution filter" if Equation (6) approaches to zero [59]:

$$d_{KL}\left(d_g \parallel d_x\right) = -\sum_{i=1}^{m} d_g \log\left(\frac{d_g}{d_x}\right) = 0 \tag{6}$$

where $d_{KL}$ denotes the Kullback–Leibler divergence of filter distributions $d_g$ and input distributions $d_x$.

CNNs use the backpropagation technique for their cost function optimization, and the filter size totally depends on the dataset features. For the ATRP, $g_i(\theta)$ with dimensions of $5 \times 1$ is suggested, as well as our DNN filter size. Our proposed CNN architecture, adapted to the ATRP upon conducting various experiments, is shown in Figure 3, which has only one difference with respect to Figure 2. In this architecture, there are convolution layers followed by a max-pooling operation which interpolates the dimensions of the outputs to half of it for each dense layer. This operation reduces potential noise in the input vectors.

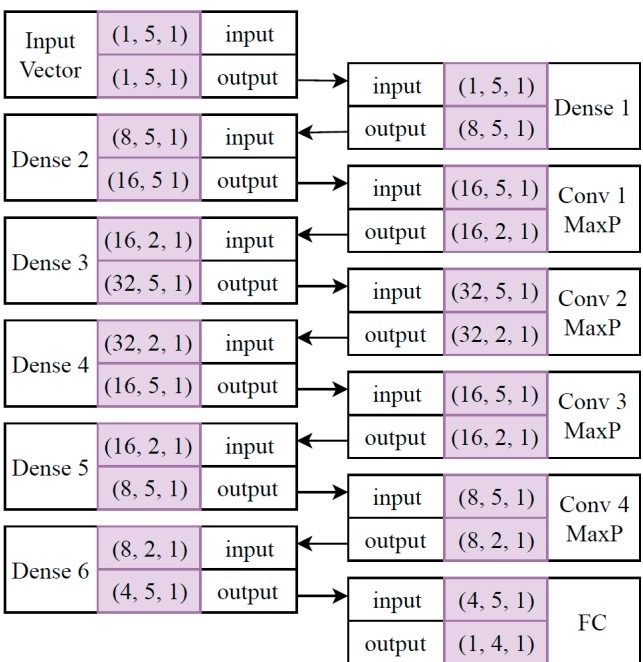

**Figure 3.** Proposed convolutional neural network (CNN) architecture for the ATRP.

The initialization scheme, batch normalization, regularization, and dropout ratio have been set to be the same as the ones of our DNN architecture, with one exception: the inclusion of a max-pooling (MaxP) operation for dimension reduction and noise removal purposes. Max-pooling is a post-processing operation that usually comes after convolution layer operation, which shrinks the input dimensions by half. It has been shown that these operations reduce potential noise in the input vectors. The ratio of MaxP is chosen to be 0.5 in order to reduce the input sequence by half, which is a default value for all the deep learning packages.

Although this proposed CNN architecture outperforms the aforementioned DNN, it is still not appropriate for our regression task. Therefore, this architecture is extended to include some recurrent blocks, thus aiming to improve the characterization of input sample distributions.

### 3.5. Recurrent CNN (RNN)

Recurrent neural networks (RNN) are versions of CNNs/RNNs developed for complex input streams (input data distributed over time characterized by strong dependency between consecutive vectors) as their current states are dependent upon their previous and subsequent states. In fact, multiple feedbacks among the layers are needed to maintain the dependency of distributions. For this type of dataset, a recurrent neural network (RNN) may outperform common CNN. RNN implements transitions between consecutive input vectors, which are distributed over time using connected states. Each state is similar to a hidden layer in a typical CNN. Connection between states could be bijective from one state to another, which is called "feedback". RNN states and feedbacks can extract the input vector dependence on time.

Since samples in our dataset are distributed over time, a suitable RNN architecture is proposed for the ATRP, as shown in Figure 4. The interpretation of this architecture is as same as the one shown as Figure 3 with one difference. In this architecture, feedbacks from one state (layer) to another state have been shown by dotted arrows. For example, gradient information which has been computed for states of Dense 5 will be transferred to states of Dense 3, and affect its weight vectors. This setup tracks temporal dependency in a sequence of input streams.

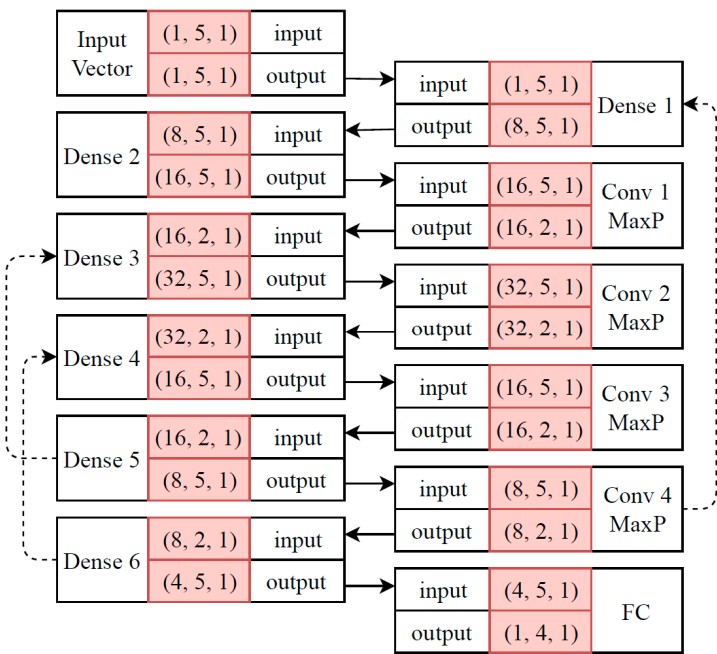

**Figure 4.** Proposed recurrent CNN (RNN) architecture for the ATRP.

This architecture is similar to our designed CNN-based architecture. Hidden layers have been empirically designed to achieve competitive performance with our CNN. The activation function used for this RNN is "tanh with a random bias vector". Kernel, bias, and recurrent initializers have been set to a truncated normal distribution of samples with $\mu = 0.5$ and $std = 0.5$. The constraints applied to the recurrent blocks are max-norm, while there are no constraints defined for kernel or bias. As explained earlier, the dropout is a regularization scheme which randomly freezes some weight vectors from their updates. The dropout ratio identifies the probability of randomly selection of neurons. When this dropout ratio is set to 0.5, then it means that there is a 50% chance for every neuron to be frozen in each epoch. In this paper, a dropout ratio of 0.5 was considered for all the layers. There are three recurrent blocks in our proposed RNN.

Since there is no optimal approach in order to automatize this process, different feedbacks have been tested for hidden layers. While designing the architecture for our RNN, consecutive feedbacks (from one state to another) were discovered from one hidden layer to another in addition to gradient

saturation to memorize their dependencies among samples, in which the generalizability of the model was negatively affected. In other words, the chance of overtraining of RNN is very high, which highly depends on the number of feedbacks among hidden layers. If this setup is not tuned properly, the gradient information might be saturated, and the training track would be lost. Therefore, a feedback was set up from the sixth dense layer to the fourth and from the fifth to the third dense layer.

Since the dimensionality of our input training data is low, $5 \times 1$, a connection is set from the fourth convolution layer to the first dense layer in order to rectify the gradient vanishing problem. "Gradient vanishing" refers to any operation which may give a "zero" value to the gradient information. If gradient vectors vanish, then no weight vector can be updated. As shown in our previous networks, the last layer of the RNN's final mapping is fully connected to the output.

Although RNNs are very much qualified for time-distributed feature learning, some short-term dependencies of the input vector among its measured states may be lost within training. Short-term dependency can be expressed by the relation between velocity and acceleration, or velocity and displacement. Therefore, a long short-term memory (LSTM) algorithm is implemented to solve the problem discussed here.

### 3.6. Long Short-Term Memory (LSTM)

Currently, the use of LSTM algorithms represents the cutting-edge data-driven approach for classification tasks as they are conveniently generalizable for regression problems. LSTMs incorporate three major gates: input, output, and forget [60]. The LSTM relying on cooperation of these blocks can temporarily remember some information about previously input vectors. The forget gate is used for tracking similar patterns over time/sequence. The schematic of our LSTM adapted to our regression task is depicted in Figure 5. Dimensions of our LSTM input are $5 \times 1$ for a given number of five measured parameters corresponding to the latitude, longitude, altitude, speed, and time for input vector. The dimensions of our LSTM output are $4 \times 1$, and include four predicted parameters corresponding to the latitude, longitude, altitude, and time for output vector. Gates have been represented by blue blocks followed by hyperbolic operations of $h$. This module tunes the timing of output vectors derived from the states.

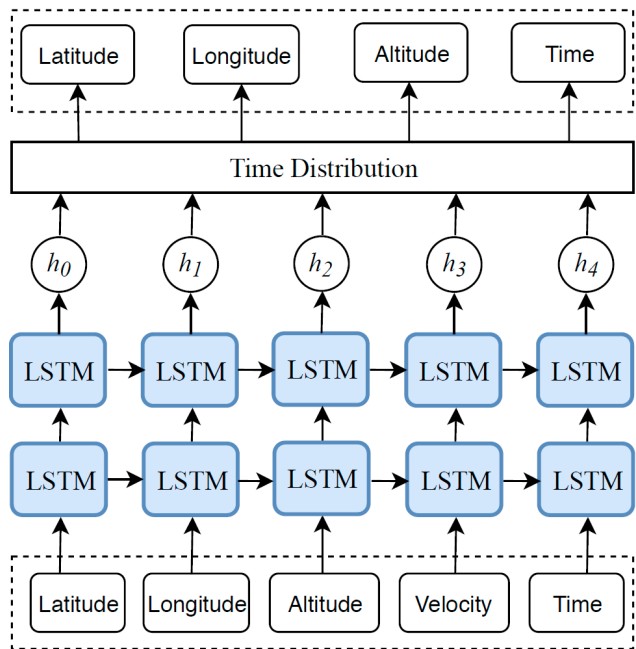

**Figure 5.** Proposed long short-term memory (LSTM) architecture for the ATRP.

As seen in Figure 5, the conducted LSTM architecture includes the input unit followed by a hyperbolic tangent (tan h) activation function. For recurrent activation, a "hard sigmoid" (between LSTM blocks) is used, through the linear activation function works fine as well. Kernel, recurrent, and bias initializers are chosen to be "random uniform functions" with "min" and "max" values equal to 0.05. For all these modules, $L_2$ similarity metric (Euclidean distance) is embedded with regularization constant $\lambda = 0.01$. For simplicity of calculations, we have not taken any kernel, bias, or recurrent constraints into account. Moreover, the dropout ratio for all the layers was tentatively set to 0.5. No sensitivity analysis was conducted for this LSTM architecture. We have adopted this network due to its very good regression accuracy with respect to the accuracy of previous deep networks.

## 4. Numerical Results

In this section, a brief explanation is provided regarding the dataset used and utilization procedure, and then the prediction results of several models are discussed.

### 4.1. Dataset

The benchmarking dataset for conducting the proposed research is the traffic flow management system publicly available online for educational use. Each record of this dataset contains latitude, longitude, altitude, velocity, and time obtained from 1676 flights [61].

### 4.2. Measuring the Resiliency of Models

As stated earlier, all six employed algorithms were trained with the maximum generalizability possible for our dataset. We implemented a 10-fold cross validation [62] for all these models and then controlled their comprehensiveness by using an early stopping technique. Experiments were further performed with the aim to determine the extent to which these models could resist given perturbations and random noise.

We assume that the trained model was built, including post-activation operations, on the training set of $x_i \in \{X\}$. The following optimization problem was further solved:

$$\min_{\epsilon} \parallel x - \widetilde{x} \parallel \leq \epsilon \ \ s.t. \ f(x_i) \neq f(\widetilde{x}). \tag{7}$$

In general, this optimization problem is known as an "adversarial attack" [63], which results in producing samples similar to the original samples but which might lead to mistakes in the model and, thus, its correction is needed.

Although classification and regression tasks are similar to each other, Equation (7) should be updated to non-label values for regression problems. In fact, unlike classification, there is no label for input vectors in the regression task. Therefore, to justify adversarial optimization problem we would need to replace the "label" with a "threshold" and solve for it. To achieve the minimum perturbation of $\epsilon$, the following optimization statement is suggested:

$$\min_{\epsilon} \parallel x - \widetilde{x} \parallel \leq \epsilon \ \ s.t. \ \min_{\delta} \parallel f(x_i) - f(\widetilde{x}) \parallel \geq \delta. \tag{8}$$

Optimizing for $\epsilon$ and $\delta$ generates a series of samples that are remarkably similar to the legitimate inputs, but they are totally different to their associated outputs. In other words, after running the optimization inequality as defined in Equation (8), the manipulated input, $\widetilde{x}$, is similar to the given legitimate input $x$ although their associated output vectors are not similar. This optimization problem could be developed to include certain conditions, namely by redirecting the $f(\widetilde{x})$ towards a predefined or random value, which can identify a targeted attack. This condition could add overhead to our abovementioned optimization problem and, therefore, we do not analyze it in the current paper. In our future studies, we will study possible approaches for the defense of our developed prediction models against adversarial attacks.

Having access to the training set, parameters, and hyperparameters of the trained model constitute a white-box attack, although it would still be possible to attack even without them. Both white- and black-box attacks are explained next.

The architectures and training setups of all six models were all the same in this paper, as explained earlier. For the training data with columns of latitude, longitude, altitude, time, and speed, the models were finely trained to predict their future states (latitude, longitude, altitude, time). The given input sample $x_i$ was randomly perturbed while keeping it close to its associated original value by using an $L_2$ similarity metrics. There is no generic approach to define the exact values for these hyperparameters. We empirically obtained these values and they can be changed following the adversary's suggestions. Here, the initial values assigned to $\epsilon$ and $\delta$ are 0.01 and 100, respectively. Table 1 summarizes the values of $\epsilon$ and $\delta$ achieved for all models trained on the traffic flow management system (TFMS) public dataset of aircraft trajectories.

**Table 1.** Mean values of $\epsilon$ and $\delta$ for training samples of the TFMS dataset.

|  | LR | SVR | DNN | CNN | RNN | LSTM |
|---|---|---|---|---|---|---|
| $\epsilon$ | 0.0103 | 0.0174 | 0.0139 | 0.0165 | 0.0237 | 0.0142 |
| $\delta$ | 59 | 126 | 207 | 67 | 106 | 92 |

Table 1 compares $\epsilon$ and $\delta$ values found by use of six benchmarking regression algorithms. Basically, adoption of smaller values for $\epsilon$ results in higher similarity between generated adversarial samples and their associated legitimate samples. Additionally, adoption higher values for $\delta$ leads to higher discrepancies between the ground-truth and the predicted outputs. Ground-truth is defined for supervised learning methods in order to measure the accuracy of the training set. Among these models, higher values for $\delta$ were achieved using DNN, which means this model yields higher variation in its predictions for legitimate inputs.

We generated adversarial samples for all the records of the dataset and we tested them by using of all the trained models. Interestingly, by applying these samples, all models predicted incorrectly.

Table 2 lists the fooling rates of all six models with their prediction confidence scores. This table compares fooling rates of six victim models against adversarial attacks that were generated by FGSM algorithm. Unfortunately, all these models were completely vulnerable against adversarial samples. The results shown in Table 2 clearly restate a security concern regarding the robustness of the data-driven models, including the conventional and advanced deep learning architectures. Scaled values of prediction confidence reveal the weakness of each model in terms of its prediction. The main difference between these algorithms is their prediction confidence. Apparently, RNN predicted wrongly with the highest confidence.

**Table 2.** Fooling rate and prediction confidence of the models.

|  | LR | SVR | DNN | CNN | RNN | LSTM |
|---|---|---|---|---|---|---|
| Fooling rate | 100 | 100 | 100 | 100 | 100 | 100 |
| Prediction confidence score | 0.784 | 0.843 | 0.732 | 0.879 | 0.910 | 0.881 |

Another important concern is the transferability of the generated fake samples from one model to another. To evaluate this situation, adversarial samples were crafted for each model, and were feed-forwarded to another model. The results of this experiment are shown in Table 3. This table statistically explains the transferability property of adversarial samples.

**Table 3.** Transferability of adversarial samples from one model to another model.

|      | LR    | SVR   | DNN   | CNN   | RNN   | LSTM  |
|------|-------|-------|-------|-------|-------|-------|
| **LR**   | 100   | 78.36 | 84.14 | **91.23** | 89.66 | 91.17 |
| **SVR**  | 81.23 | 100   | 84.17 | **95.07** | 84.56 | 89.59 |
| **DNN**  | 90.07 | 89.23 | 100   | 95.81 | **97.33** | 94.46 |
| **CNN**  | 86.75 | 88.71 | 91.63 | 100   | 91.55 | **93.57** |
| **RNN**  | 97.29 | 94.58 | 90.67 | 95.58 | 100   | **98.26** |
| **LSTM** | 79.16 | 81.92 | 89.99 | **93.52** | 88.37 | 100   |

This table depicts the transferability of adversarial samples from one victim model to another. Reported percentage values are averaged among all 10 folds, which is equivalent to say that the given dataset was divided into 10 equal-size segments versus time and, thus, each one of them was considered a test segment. Finally, the average of accuracy was we computed for these segments. The most transferable adversarial samples for each model are shown in Table 3 in bold characters. For instance, 81.23% of total crafted adversarial samples for SVR are successfully transferable to the LR model.

Although LSTM is more advanced than the RNN, it is more vulnerable to transferred adversarial attacks. Equation (8) is further explored for a better understanding of crafted samples. A first impression could be that adversarial samples are "noises". To accept or reject this impression, we need to run experiments to determine if $\epsilon$ and $\delta$ constitute "noise" (or not).

To answer the abovementioned question, we utilized the local intrinsic dimensionality (LID) score [64]. This score differentiates "noisy samples" from "crafted adversarial samples". Assuming that $d_i(x)$ refers to the distance from legitimate sample $x_i$ to its nearest neighbors, $d_k(x)$, then the maximum of the neighbor distances can be found in which $k$ is the number of neighbor samples. Therefore, the LID score can be computed as shown in Equation (9).

$$LID(x) = -\left( \frac{1}{k} \sum_{i=1}^{k} log \frac{d_i(x)}{d_k(x)} \right)^{-1} \tag{9}$$

Around 15% of the training set and generated random noisy samples were randomly selected using Gaussian distribution with 10 different values of $\mu \in [-1, 1]$ and $\sigma \in [-0.75, 0.75]$. For fairness comparison, we repeated this generation 10 times and exported all the generated noisy samples into the original dataset by building a new directory to include both noisy and legitimate samples. We also generated new adversarial samples for every record in the original training set and further exported them into the adversarial category. Eventually, a logistic regression algorithm is trained for two considered classes in order to classify legitimate and adversarial samples. Table 4 summarizes the details of this binary classification.

**Table 4.** Performance comparison of logistic regression (LR) on local intrinsic dimensionality (LID) scores. The solver for this LR classifier is "liblinear".

|  | Max Iteration | Training Accuracy (%) | Test Accuracy (%) | Penalty | Tolerance | Fitting Intercept | # Jobs | C |
|---|---|---|---|---|---|---|---|---|
| Without cross-validation | 120 | 86.36 | 84.27 | $L_2$ | $1e^{-5}$ | False | 4 | 0.002 |
| 5-fold cross-validation | 100 | 91.23 | 87.75 | $L_2$ | $1e^{-5}$ | False | 4 | 0.001 |
| 10-fold cross-validation | 95 | 92.13 | 86.49 | $L_1$ | $1e^{-6}$ | True | 8 | 0.003 |
| 15-fold cross-validation | 85 | 92.67 | 86.18 | $L_1$ | $1e^{-6}$ | True | 8 | 0.002 |

Table 4 primarily compares the accuracy of LR on the LID scores as well its setups for training. For example, the first row of this table shows that LR without cross validation has 86.36% and 84.27% accuracy in training and test, respectively. These accuracies have been achieved at the 120th iteration with $L_2$ regularization penalty and with a prediction tolerance (error) of $1e^{-5}$. Training has been executed using four CPU core (jobs) without weight normalization (false fitting intercept). The inverse of the regularization strength (C) for this model is set to 0.002.

As shown in Table 4, the LR is favorably used for the binary classes of the LID scores, and it supports our previous hypothesis (can adversarial samples be interpreted as noisy samples or not?) regarding the fundamental difference between noisy and adversarial samples. For a very good characterization of the distribution values of the original, noisy, and adversarial samples, we plotted their LID scores in Cartesian space. Please note that LID is a score given to every input. Figure 6 visually shows distribution of LID scores for triplet of original, noisy, and adversarial samples.

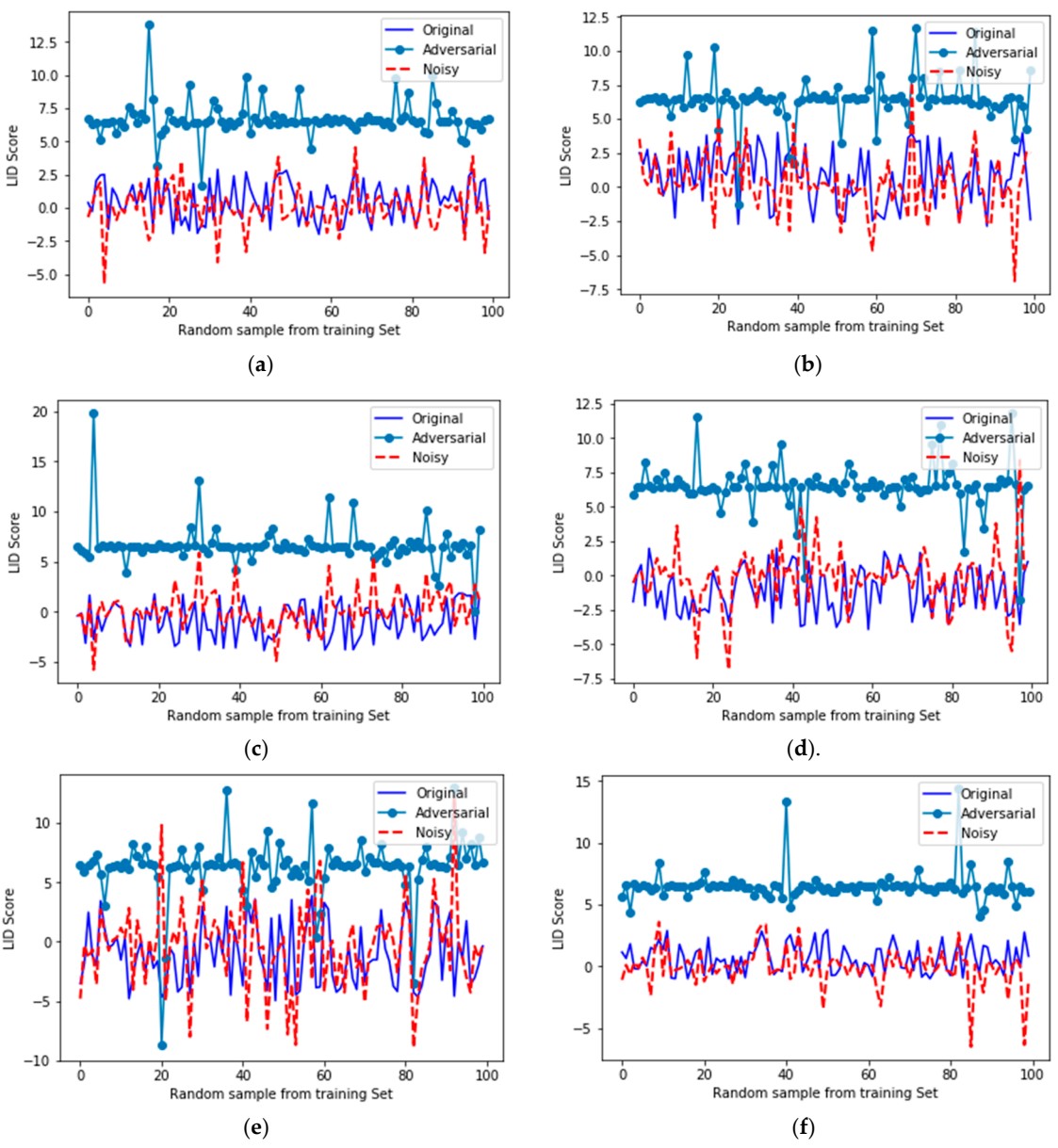

**Figure 6.** LID score comparisons for different random samples chosen from the training set. (**a**) The first random set; (**b**) The second random set; (**c**) The third random set; (**d**) The fourth random set; (**e**) The fifth random set; (**f**) The sixth random set.

Figure 6 shows the LID score comparisons for random samples chosen from the training set. As this figure indicates, original and noisy samples lie in the same LID subspace, which denotes their structural similarity. Conversely, adversarial samples are located in a separated upper subspace different from the original and noisy sets. To demonstrate that these LID scores were also statistically different, we trained an LR in order to classify LID scores of original, noisy, and adversarial samples. Obviously, higher values of accuracy of the trained LR mean better classification for LIDs. We summarize the details of the LR in Table 4 as well as other training information.

Overall, Table 4 statistically proves that LIDs for adversarial samples are far from original and noisy samples, and Figure 6 shows this difference visually.

Generating adversarial samples with respect to the intrinsic characteristics of the given dataset could be very costly in terms of optimization overhead. In other words, Equation (8) does not always show a complex optimization task and could be a non-polynomial problem. These problems cannot be solved by polynomial functions approximation (of any degree). Therefore, Equation (8) could be replaced by a faster operation, namely, by taking advantage of gradient information backpropagated through the network during its training. Generating adversarial samples relying on gradient information was first introduced in the computer vision community, and was called "fast gradient sign method" (FGSM) [63]. We will adapt this attack for our regression task.

The FGSM is categorized as a white-box and non-targeted adversarial attack, mainly for architectures trained by backpropagation, and requires the model gradient information. For a given input $x_i$, the FGSM crafts adversarial sample $\widetilde{x}$, as defined in Equation (10):

$$\acute{x} = x + \epsilon \times sign(\nabla_x J(\theta, x, l)) \tag{10}$$

where $J$ is the cost function of the model, and $\epsilon$ is a float scalar to be defined by a local search. Since the FGSM attack was introduced for classification purposes, we needed to update the label index of $l$ to a bounded value by providing a "supremum" and an "infimum". Therefore, Equation (10) should be written under the following form [63]:

$$\acute{x} = x + \epsilon \times sign(\nabla_x J(\theta, x, v)) \tag{11}$$

where $v \notin [a - \lambda, \ a + \lambda]$ is an output value, and $a$ is the actual value as defined in the training set. Our adapted version of the FGSM (AFGSM) requires its optimization for both $\epsilon$ and $\lambda$.

In our next experiment, we generated adversarial samples using the AFGSM for our proposed DNN, CNN, RNN, and LSTM architectures. We also studied the transferability property of crafted samples, as shown in Table 5. Table 5 compares the transferability of adversarial samples using our proposed AFGSM algorithm. For instance, the first element in Table 5 suggests that 78.25% of total crafted adversarial samples are successfully transferable from DNN to the LR model.

**Table 5.** Transferability of adversarial samples crafted by the adapted fast gradient sign method (AFGSM). The highest values are in bold characters.

|          | LR    | SVR   | DNN   | CNN   | RNN   | LSTM  |
|----------|-------|-------|-------|-------|-------|-------|
| **DNN**  | 78.25 | 86.94 | 100   | **91.25** | 89.36 | 90.71 |
| **CNN**  | 85.13 | 84.58 | **92.47** | 100   | 91.55 | 92.05 |
| **RNN**  | 90.96 | 92.37 | 88.24 | **93.37** | 100   | 91.08 |
| **LSTM** | 91.45 | 90.33 | 89.69 | 89.99 | **92.28** | 100   |

As shown in Table 5, all the models are vulnerable to our version of FGSM attack. Not surprisingly, generated adversarial samples using the AFGSM for DNN and CNN are the most transferable samples to each other and are shown in bold characters (91.25, 92.47). Moreover, AFGSM-generated adversarial samples for RNN architecture are the samples most transferable to the CNN model (93.37).

One hypothesis could be that this is related to their same utilized convolution layers, regardless of their filters shape, sizes, or order.

*4.3. Adversarial Retraining*

One potential defense against the threat of adversarial attack would be to train models by use of a combination of legitimate and adversarial samples. In other words, both original and crafted adversarial samples could be fed with the correct labels to the model within it training, with the aim of avoiding being misled during the testing time. Equation (12) shows our proposed retraining policy:

$$\widetilde{J}(x_i, y_i, \theta) = cJ(x_i, y_i, \theta) + (1 - c)cJ(\widetilde{x}_i, y_i, \theta) \tag{12}$$

where $c$ is a constant value set to 0.25, 0.5 and 0.75 for our dataset. Table 6 presents the performance of the retraining policy for 3 different $c$ values.

**Table 6.** Performance comparison of data-driven models by adversarial retraining.

|  | c | LR | SVR | DNN | CNN | RNN | LSTM |
|---|---|---|---|---|---|---|---|
|  | 0.25 | 86.45 | 77.36 | 80.35 | 81.06 | 78.37 | 79.08 |
| Fooling rate (%) | 0.50 | 83.25 | 76.28 | 79.47 | 81.69 | 79.84 | 74.41 |
|  | 0.75 | 84.27 | 75.79 | 80.23 | 81.97 | 80.56 | 73.19 |
|  | 0.25 | 66.16 | 56.33 | 61.74 | 57.19 | 59.67 | 61.54 |
| Regression accuracy (%) | 0.50 | 64.87 | 57.13 | 64.24 | 58.36 | 58.14 | 60.79 |
|  | 0.75 | 68.97 | 52.87 | 61.58 | 60.76 | 59.42 | 59.45 |

## 5. Conclusions

In this paper, the accuracy of data-driven regressors was investigated for conventional (LR and SVR) and state-of-the-art (DNN, CNN, RNN, and LSTM) algorithms for aircraft trajectory prediction by use of the traffic flow management system (TFMS) of aircraft trajectories. Although the results testify the higher performance of the modern algorithms in terms of regression accuracy, they also show the lowest resiliency against crafted adversarial attacks. We implemented FGSM and AFGSM adversarial attacks for all the trained models, and measured their fooling rates. Interestingly, conventional classifiers showed a higher robustness to adversarial attacks compared to the advanced deep neural networks. As a pro-active approach for improving the robustness of the models, we adversarially trained all of them, which also increased their error rates. This increased error rate poses a security issue for learning-based regressors, especially since adversarial samples are transferable from any learned model to another model, as already shown. For our future work, a data-driven regression algorithm will be developed that will give a reasonable tradeoff between regression accuracy and fooling rate.

**Author Contributions:** Conceptualization, S.M.H.; Data curation, S.M.H.; Funding acquisition, R.M.B.; Investigation, T.L.G.; Methodology, S.M.H. and T.L.G.; Project administration, R.M.B.; Resources, R.M.B.; Software, R.M.B.; Supervision, R.M.B. All authors have read and agreed to the published version of the manuscript.

**Funding:** This research was funded by NSERC within the Canada Research Chairs program, which made possible the realization of this research and the publication of this paper.

**Conflicts of Interest:** The authors declare no conflict of interest. The funders had no role in the design of the study; in the collection, analyses, or interpretation of data; in the writing of the manuscript, or in the decision to publish the results.

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
