# Peer review of "New Reliability Studies of Data-Driven Aircraft Trajectory Prediction"

_aerospace, doi:10.3390/aerospace7100145_

Round 1

Reviewer 1 Report

This work present the aircraft trajectory prediction by using 6 different algorithms  and compared results.The paper is well-written on the topical area and a further comment is given here: 

Some numbers in the tables is better to be shown in one row, such as C in Table. 4.

Author Response

Authors’ answers to reviewers’ comments for journal paper entitled

Reliability New Studies of Data-driven-based Aircraft Trajectory Prediction

The authors would like to thank very much to reviewer 1 and to the editor for their comments, which helped them to improve the writing of this paper.

Comments of Reviewer 1:

This work presents the aircraft trajectory prediction by using 6 different algorithms and compared results. The paper is well-written on the topical area and a further comment is given here:

Some numbers in the tables is better to be shown in one row, such as C in Table 4.

Authors’ Answers to the Comments of Reviewer 1:

The numbers in Table 4 are shown in one row.

Many thanks once again for the revisions, and in advance for the revised paper, the authors

Reviewer 2 Report

The presented article is highly theoretical well prepared. I can consider the evaluation of accurate data to be original. However, what I miss the most in the article are the ambiguities in the introduction by describing one model approach of aircraft in fig. 1, which is generally a normal approach of aircraft and solvable. Such a convergence of aircraft is certainly solved by the TCAS system you know. In the introduction, therefore, I lack the purposefulness of the article, what and where exactly should be changed on which avionic system, whether it is already on the aircraft, or on the evaluation ground control system of flight control. The complexity of the calculations clearly indicates your theoretical skills in controlling the issue, but it would be appropriate and provable to approximate the real facts arising from the ratio and acceleration of, for example, signalling and automation of avoiding a pair of aircraft.
In the article, you often describe a model of an aircraft, which can be described or at least approximated by what model it is or how it is mathematically created by at least a simple differential equation of its direct motion.
In paragraph (89), I do not understand the use of the UAV control system. Please check the introductory sections on the use of aircraft collision avoidance systems and what your research could be improved in your resources.

Author Response

Authors’ answers to reviewers’ comments for journal paper entitled

Reliability New Studies of Data-driven-based Aircraft Trajectory Prediction

The authors would like to thank very much to reviewer 2 and to the editor for their comments, which helped them to improve the writing of this paper.

Comments of Reviewer 2:

The presented article is highly theoretical well prepared. I can consider the evaluation of accurate data to be original.  However, what I miss the most in the article are the ambiguities in the introduction by describing one model approach of aircraft in fig. 1, which is generally a normal approach of aircraft and solvable. Such a convergence of aircraft is certainly solved by the TCAS system you know. In the introduction, therefore, I lack the purposefulness of the article, what and where exactly should be changed on which avionic system, whether it is already on the aircraft, or on the evaluation ground control system of flight control. The complexity of the calculations clearly indicates your theoretical skills in controlling the issue, but it would be appropriate and provable to approximate the real facts arising from the ratio and acceleration of, for example, signalling and automation of avoiding a pair of aircraft.

In the article, you often describe a model of an aircraft, which can be described or at least approximated by what model it is or how it is mathematically created by at least a simple differential equation of its direct motion.

In paragraph (89), I do not understand the use of the UAV control system. Please check the introductory sections on the use of aircraft collision avoidance systems and what your research could be improved in your resources.

Answers to Comments of Reviewer 2:

The main aim of this article is the fixed-wing aircraft trajectory prediction in a specific root which can be considered as an excellent tool for a variety of aerial operations. This prediction may be performed for different time duration namely, short-term and long-term. The long-term prediction is useful for air traffic control, fuel consumption optimization, and logistic operations while the short-term prediction is useful for conflict detection. The predicted trajectories may be utilized on the ground computer unit of the Air Traffic Control System (ATCS) or computer unit of the Aerial Collision Avoidance System (ACAS) in cockpit.

The above paragraph was introduced on pages 1 of Section 1 of the revised paper in blue color. This paper is focused on aircraft trajectory prediction accuracy and reliability that is the first objective of my PhD thesis. The accuracy of predicted trajectories was assessed using the error rate in the test phase, and the reliability of ATP neural network models were evaluated based on the fooling rate for the adversarial attack. The ACAS performance analysis based on data-driven trajectory predictors is not the aim of this article. The next objective of my PhD thesis will concern the ACAS performance analysis for aircraft encounter situation while novel defence techniques against adversarial attacks will be employed.

The above paragraph was introduced on pages 2 of Section 1 of the revised paper in blue color.

Although encounter scenarios as the one shown in Figure 1 were solved using TCAS (without future trajectories), relying on predicted trajectories the collision avoidance task can be performed more optimal. Actually, it is known that the TCAS modifies the owner future trajectory if an intruder enters into the owner resolution advisory zone. Moreover, the TCAS design is based on the current aircraft position, and on its conservative virtual unsafe zone. Hence, many false alarms and unnecessary resolution advisory may occur during the flight. In this way, the collision zone can be reduced using an accurately predicted trajectory (position of aircraft in the th step), which leads to avoiding unnecessary trajectory modification. Therefore, the design of a reliable and precise trajectory prediction algorithm is needed.

The above paragraph was introduced on pages 2 and 3 of Section 2 of the revised paper in blue color.

The trajectory may be predicted using conventional approaches (e.g. Model Predictive Control MPC) based on aircraft dynamics model or modern data-driven techniques (e.g. Deep Neural Network DNN) relying on large amounts of recorded data. Studies have shown that modern data-driven techniques outperform conventional approaches if enough training data is available, and security issues are respected. It is known that in conventional approaches, uncertainties back-propagate through the prediction horizon, and error increase dramatically. Hence, the data-driven algorithms were adopted for trajectory prediction, and the ATP task was performed regardless of aircraft dynamics model, which is a remarkable advantage of data-driven predictors.

The above paragraph was introduced on page 4 of Section 2 of the revised paper in blue color.

In the literature review on conventional approaches in the introduction section, sometimes the “aircraft model” was mentioned. But the mentioned “model” in this paper refers to the neural network model that was supposed to predict aircraft future trajectories. As explained earlier, the proposed methodologies with the aim to build data-driven predictors do not require an aircraft dynamics model, and these predictions were performed using trained neural networks (the so-called model in this paper).

In line 103 (line 89 before editing), when an intruder arrives in the vicinity of the owner’s neighborhood, after intruder detection, ACAS resolution advisories commands should be transferred to the fixed-wing aircraft control system, which is supposed to deflect control surfaces with the aim to modify future trajectory. The control system of the owner aircraft (that is flying in a specific root) updates its subsequent trajectory.

Many thanks once again for the revisions, and in advance for the acceptation of this revised paper, the authors

Round 2

Reviewer 2 Report

The authors extended the issue with recognized understandings for professional readers in the field of anti-collision systems. The authors modified the article so that it is clear what is the problem of onboard aircraft and ground colision monitoring systems in the field of ACAS. The attached simulations show a predictive system of improvement of these systems if it were the issue of air traffic compaction described in the article.